# Information Sharing and Investment Performance in the Venture Capital Network Community: An Empirical Study of Environmental-Social-Governance Start-Ups

**DOI:** 10.3390/ijerph16061023

**Published:** 2019-03-20

**Authors:** Chaokai Xue, Xinghua Dang, Beibei Shi, Jing Gu

**Affiliations:** 1School of Economics and Management, Xi’an University of Technology, Xi’an 710054, China; xuechaokai@gmail.com (C.X.); gsg1001@xaut.edu.cn (X.D.); 2School of International Trade and Economics, University of International Business and Economics, Beijing 100029, China; shib_123@163.com; 3School of Economics, Sichuan University, Chengdu 610064, China

**Keywords:** environmental, social and governance (ESG), venture capital, network community, information sharing, investment performance

## Abstract

Information sharing plays a significant role in improving investment performance in the venture capital network community, which serves as an important requirement for the venture capital network to expand to the meso-level and promote its healthy development. Given the important role of Environmental-Social-Governance (ESG) start-ups in promoting sustainable development, this paper selects a sample of ESG start-ups in China to explore the relationship between venture capital network community information sharing and investment performance. We also examine the intermediary role of investment capability in this relationship. Empirical results show that venture capital network community information sharing, from both the prospective of breadth and depth, has a significant positive impact on investment performance of ESG start-ups. We also find that the investment capability, such as scouting and coaching, plays a partial intermediary role in affecting investment performance by community information sharing. This research helps to reveal the relationship between venture capital network community information sharing and investment performance. The results provide a theoretical basis and managerial insights for improving investment performance of ESG start-ups.

## 1. Introduction

Owing to increasing awareness and promotion of sustainable development, environmental, social and governance (ESG) issues are gaining prominence in comprehensively evaluating responsible investing activities. It is confirmed by Margolis and Walsh [1] that ESG issues can have a major role in generating economic values from financial investments and in the development of different nations [2]. According to the 2016 GSIA’s report [3], ESG investment scale is growing at a rapid pace. For example, Europe and the United States account for 90 percent of the global ESG investment market.

ESG also refers to extra financial material information about the challenge and performance of organizations and requires investors to act in the best long-term interests of the society and beneficiaries [4]. Eurosif [5] further asserts that Venture Capital (VC) investors can influence start-ups toward sustainability by integrating ESG issues into investment decisions and client relations. As is well-known, VC plays an important role in promoting the development of start-ups over an extended period including the pre-investment and post-investment stages. In the pre-investment stage of VC, attention should be given to effective ESG due diligence [6]. During the postinvestment management of start-ups, Venture Capitalists (VCs) always share the information of start-ups with one another, including ESG issues in a VC network [7]. Actually, due to the high risk of investing in start-ups, a syndicated investment strategy is often adopted by VC to reduce risk and share resources, thereby forming a VC network.

Several studies have confirmed that the VC network is the pipes and prisms of information sharing [8], which has an important impact on the investment behavior and performance of the VC [9,10], such as the investment region and investment industry [11]. However, previous studies in the field of VCs only mention the role of information sharing from the perspective of a theoretical analysis; very few studies have made an in-depth analysis by empirical methods.

It has been revealed that the VC network has a tendency for information sharing, and the main information that is shared in the network includes start-ups’ ESG information and investment experience information. This paper argues that community information sharing, which is one of the direct means of project screening and value-added services, will have an important impact on investment performance. In addition, the information shared by the community is also the micro-foundation of the construction of the community investment capacity. The breadth of information sharing affects the richness and diversity of resource acquisition. The depth of information sharing affects resource allocation and resource reorganization and then influences the scouting and coaching capability of the community, which ultimately has an impact on investment performance. Therefore, this paper confirms that community information sharing can also indirectly affect investment performance through community investment capability.

Although previous studies have confirmed that the VC network has the function of information sharing and has an impact on the VC’s investment behavior and performance, there are few empirical studies taking information sharing as an independent variable. Whether and how information sharing affects investment performance in the context of the network community has not been thoroughly elucidated to date. This paper examines the impact of information sharing on investment performance and further explores the influence mechanism of community investment capacity as an intermediary variable. By using a sample of ESG start-ups, this study helps to understand the role of community information sharing on investment performance, and provides a theoretical basis and new angle for improving investment performance.

The contributions of this paper are mainly reflected in the following aspects: First, existing research on VC networks mainly focus on the micro-self-central network level and the macro-integrated network level. However, as limited research has been conducted from the perspective of the network community, this paper examines the impact of VC network community information sharing on investment performance, which can enrich and expand the theory of the VC network from the perspective of the network community. Second, the research enriches and expands the factors affecting investment performance from the perspective of community information sharing. Existing research indicates that the VC network has the function of information sharing, and but to the best of our knowledge, no research has investigated it as an independent variable. This paper studies the investment performance from the perspective of information sharing, which enriches the literature in the field of the influencing factors of investment performance. Third, this paper reveals the path of community information sharing in the VC network to influence investment performance and provides new insights for improving investment performance and helps ESG start-ups to grow sustainably.

The rest of the paper is organized as follows: Section 2 presents literature review. Section 3 provodes theoretical framework and the hypothesis. Section 4 shows research design. Section 5 demonstrates our empirical results; and the final section provides conclusions.

## 2. Literature Review

Existing studies have mentioned that the VC network has the function of information sharing, and this function has had an important impact on investment performance, but these studies have not been explored in depth and only from the perspective of theoretical analysis. In this part, we first review VC network community information sharing based on existing literature.

According to Sorenson and Stuart [11], information about potential investment opportunities usually circulates in geographic and industrial spaces, which restrict information sharing and make VCs invest locally in geography and industry. However, VC networks enable information sharing across borders and promote cross-border investment among network members. Further, Podolny [8] regards the network as the pipes and prisms of market information sharing. The VC network promotes the exchange and sharing of information and other resources, such as investment experience information in management, operation of funds under its flag, management of companies, and investment project information, such as industry prospects and future investment opportunities. In the related fields of information sharing, such as information or knowledge searching, knowledge searching is divided into two classical dimensions: breadth and depth [12,13]. Searching breadth refers to the scope and range of organizational knowledge searching, that is, the number of external knowledge sources or searching channels involved in organizational knowledge searching activities. Searching depth refers to the extent to which an organization makes deep use of external knowledge sources or search channels in the process of knowledge search.

Existing literature has noted that the VC network community provides a more favorable environment for information sharing, and there is indeed information sharing in the network community. According to Bubna et al. [14], the connection between independent companies helps to avoid the rigidity and complexity of the connection between large enterprise groups, but it is not conducive to the sharing of soft information and the integration of relationship resources, while the VC network community is a special form of organization between independent companies and large enterprise groups and can combine the advantages of independent companies and enterprise groups.

From the perspective of the VC process, Yang et al. [15] classifies VC capability as screening and evaluation capability. The screening capability is the capability to select a start-up that generates financial returns in the short term or generates a strategic return in the long run. The evaluation capability refers to the ability to assess the venture project reasonably. Bertoni et al. [16] indicates that due to excellent investigative capability, the VC can identify and provide the necessary financing for the start-ups with hidden value. At the same time, VC can actively monitor start-ups and increase their value. Meglio et al. [17] divides VC capability into scouting and coaching capability and holds that scouting skills refer to the capability developed by some VCs in identifying particularly attractive new ventures, unleashing their superior competitive potential and spurring their growth by relaxing financial constraints on the new ventures’ development, while coaching skills refer to the capability developed by some VCs to extend the new ventures’ bundle of strategic resources and capabilities during their holding period.

The existing research on VC in ESG start-ups focuses on the importance of VC for ESG start-ups funding dilemmas. However, VC in ESG start-ups is facing greater uncertainty, which is associated with greater investment risks. At this time, information about ESG start-ups is shared among the VC network community, which can alleviate this uncertainty. This can help VC improve its investment performance and contribute to the healthy development of ESG start-ups.

Margolis and Walsh [1] confirms that ESG issues have a major role in the generation of economic value from financial investments. Driven by political pressures to lower CO_2_ emissions and ESG investment philosophy, the ESG start-ups have become an important target for VC investments [18]. However, investing in ESG companies has more uncertainty and investment risks than for in other companies. Petkova et al. [19] notes that when investing in ESG start-ups, such as investment in an emerging sector, VCs not only face uncertainty about the quality and potential of a particular start-up but also lack the fundamental understanding of how to think about quality and potential in the sector. Therefore, information sharing about ESG start-ups in the VC network community is particularly important.

## 3. Theoretical Framework and Hypothesis

### 3.1. Community Information Sharing and Investment Performance

First, information asymmetry is a key factor affecting investment performance. In the field of VCs, the problem of adverse selection of hidden information refers to the investment in a “lemon project” by venture capital. The moral hazard problem caused by information asymmetry refers to entrepreneurs’ burnout or insufficient efforts. Therefore, it is a key path for VCs to improve investment performance by collecting information on ESG start-ups before investment and strengthening supervision and control of ESG start-ups after investment. Second, the VC network has the function of information sharing, which can alleviate the information asymmetry in the investment process. To diversify risk and share resources, VCs often make syndicated investments and form a VC network. The VCs can use the information sharing function of the VC network to collect and obtain the quality information before the investment and the business management status information after the investment to alleviate the information asymmetry in the ESG start-ups investment process. Third, the VC network community provides better environmental conditions for project information sharing and experience information sharing and will have an important impact on investment performance. The VC network community is a subnetwork with close internal connections and sparse external connections formed by syndicated investment. It has a characteristic cohesive feature and is a typical strong joint subnetwork. Therefore, the structural cohesive characteristics of the VC network community provide convenient external conditions for information sharing, expand the scope of project selection for VCs and the intellectual capital required for value-added services, and enable the VC network community to screen higher quality ESG start-ups and better value-added services to improve investment performance.

### 3.2. Community Information Sharing and Community Investment Capability

First, the investment information about ESG start-ups shared by the VC network community is the micro foundation for the construction of community investment capability. Resource-based theory holds that firms have tangible and intangible resources that can be transformed into unique capabilities that are the source of a company’s long-lasting competitive advantage. Second, the breadth of information sharing affects the richness and diversity of resource acquisition which, in turn, affects the community’s scouting capability and coaching capability. The richness and diversity of information resources in the community will help the community identify investment opportunities and prioritize investment opportunities in the face of investment opportunities. In addition, after investing in ESG start-ups, the community is able to provide diversified value-added services for ESG start-ups, that is, the community’s coaching capability can be improved. Third, the depth of information sharing affects resource allocation and resource reorganization which, in turn, affects the community’s scouting capability and coaching capability. In-depth information sharing will fully identify resources at the community level and will optimize, allocate, and reorganize resources from an overall level such that the community has updated investment skills, which can then offer an advantage in the perception of investment opportunities and the allocation of postinvestment services.

### 3.3. Community Investment Capability and Investment Performance

First, the stronger the community’s investment capability is, the higher the investment performance is. For the network community, the stronger its community investment ability, the more the network community has mastered the scarce resources and the allocation of resources, and the more it can cope with the changes in the internal and external environment, maintain its competitive advantage, and ultimately act on investment performance. Second, the scouting capability affects the “first mover advantage” and plays a role in ESG start-ups project screening and project evaluation, which will affect the performance of community investment. For example, a network community with “first mover advantage” can obtain more information about investment opportunities, facilitate project screening, and be more likely to explore investment opportunities in new industries and early stage start-ups. Once these start-ups are successful, they will receive excess compensation. In addition, the network community with “first mover advantage” will be more proactive in accessing and utilizing resources such as information and learn from other community members about the “second opinion” of the investment project, which is beneficial to its project evaluation and will ultimately affect its investment performance. Third, the coaching capability affects “resource integration” which, in turn, affects value-added services, such as supervision and corporate governance, and ultimately affects community investment performance. The theory of resource integration emphasizes that organizations should not only have resources but also integrate and utilize resources. Community coaching capability can also restructure resources in the configuration process to create new knowledge and develop new investment opportunities and investment skills that collectively have an impact on community investment performance.

Based on the above analysis, this paper proposes that the breadth and the depth of information sharing will have a direct impact on investment performance and will also be transmitted through the scouting capability and coaching capability, which will have an indirect impact on investment performance. Overall, the theoretical model constructed in this study is shown in Figure 1:

### 3.4. Hypothesis

#### 3.4.1. Impact of Community Information Sharing on Investment Performance

Based on the research of Hwang and Lee [13] and Bubna et al. [14], this paper divides VC network community information sharing into two dimensions: information sharing breadth and information sharing depth, and the study investigates its impact on investment performance.

First, the higher the breadth of information sharing in the VC network community, the more information the community members obtain and the broader is the scope of investment ESG start-ups, which helps them to select better quality ESG start-ups and improve investment performance. Bygrave [20] suggests that VCs can learn whether start-ups are rejected by other members through the network, which will speed up the project screening of VCs. Cumming [21] found that VCs share information through syndicated investment to reduce the information asymmetry of investment start-ups and the information sharing between VCs on investment start-ups helps to reduce the possibility of investing in “lemon” start-ups. Beshears [22] found that information sharing among members of the Gulf of Mexico Oil and Gas Drilling Company help member companies find large-scale drilling, thereby improving their financial performance. Therefore, we believe that the higher the breadth of information sharing in the VC network community, the more information the community members can obtain about ESG start-ups, and the more it can help reduce the time of project screening and select higher quality ESG start-ups, thus improving investment performance.

Second, the higher the breadth of information sharing in the VC network community is, the more experience information the members of the community obtain, and the more skills they use in monitoring business management. These advantages enable the community members to provide better value-added services for ESG start-ups and improve investment performance after investment. Hopp [23] believes that industry investment experience provides legitimacy for leading VCs, and more industry experience can bring more syndicated investment. Huang et al. [24] found that if a company’s directors have experience in investment banking, the possibility of making merger decisions on the board is higher, and the M&A events show higher long-term performance. Similarly, Field and Mkrtchyan [25] also believe that companies will hire directors to gain acquisition experience that is positively related to subsequent acquisition performance. Therefore, we believe that the higher the breadth of information sharing in the VC network community, the more experience information members can obtain, and the more comprehensive value-added services can be provided to ESG start-ups, and thus, the investment performance can be improved.

Based on the above analysis, this paper believes that the breadth of information sharing in the VC network community about ESG start-ups can increase the amount of information of community members and facilitate project screening and value-added services, thereby improving investment performance. Therefore, we propose the following hypothesis:

**H1.** 
*The breadth of information sharing in the venture capital network community has a positive impact on investment performance.*


First, the deeper the VC network community information sharing is, the more frequent the information sharing among the members of the community is, and the more trust that can be generated between them, which helps to obtain “second opinion” on investment ESG start-ups and reduce opportunistic behavior. Lee [26] argues that information sharing through repeated transactions reduces information asymmetry among participants and the opportunistic behavior of partners. Therefore, we believe that the higher is the depth of information sharing in VC network community, the higher the trust among community members. This trust will help to provide real and effective “second-party opinions” for syndicated investment ESG start-ups and reduce opportunistic behavior in the process of cooperation, thereby improving the performance of community investment.

Second, when the information sharing depth of the VC network community is large, the information sharing among community members will be more frequent; thus, the relationship between members is intimate, making the community partners form reciprocal behavior. Ritter [27] found that when institutional investors pay more transaction commissions to the main underwriters of the IPO, institutional investors can obtain more IPO share placements. Ferrary [28] believes that the VC participating in the first round of investment is regarded as an internal investor who understands the private information of the start-ups in the subsequent round of financing. This study shares the private information of the start-ups with its familiar partners in exchange for the return of other VCs on the investment project. Therefore, we believe that the deeper the information sharing of VC network community is, the more frequent the reciprocal behavior among members of the community and the greater the pool of ESG start-ups resources that can be formed within the community, which will help to invest in more high-quality ESG start-ups, thereby improving the performance of community investment.

Based on the above analysis, we have the following hypothesis:

**H2.** 
*The depth of information sharing in the venture capital network community has a positive impact on investment performance.*


#### 3.4.2. Impact of Community Information Sharing on Investment Capability

The greater the breadth of information sharing is, and the greater the number of information channels is, the larger the information set collected by VC is, which helps to identify external opportunities, perceive investment opportunities and initiate investment activities earlier than other online communities. Ozmel et al. [10] shows that network-centric VCs have a wider access to information, which helps to understand the service needs of the start-ups. In the area of VCs, Sorenson and Stuart [11] finds that information about potential investment opportunities typically circulates within geographic and industrial spaces that limit the flow of information and allow VCs to make local investments in geography and industry. However, the syndicated investment network between VCs allows information to be shared across space boundaries, thereby reducing the spatial constraints on economic exchanges and affecting investment decisions such as VC investment regions and investment industries. Hopp [23] believes that new partners joining the syndicated investment network can be an important source of new knowledge or information. The new partners provide more heterogeneous knowledge and information for the syndicated investment network, which helps the members of the original investment network to expand investment areas and investment industries.

The greater the breadth of information sharing, and the more members or channels in the community for start-ups project information or postinvestment experience information, the higher is the information heterogeneity among community members, which helps the VC network community allocate resources such as information according to the characteristics of the start-ups. Henderson and Cockburn [29] argues that new heterogeneous information or knowledge is necessary for solving unconventional problems. In addition, the more information sources or channels community members have, the more complementary the information among community members. As a result, the community has a greater chance of reorganizing or utilizing a variety of information, which helps the VC community to create new information or new knowledge and to allocate resources for information such as new investment opportunities. The study by Fleming and Sorenson [30] confirms that the greater the breadth of knowledge sharing, the greater are the chances of reorganizing and integrating knowledge, which will drive new product development activities.

Therefore, we believe that the greater the breadth of information sharing, and the greater the number of information collections shared by the VC community, the more it can identify external investment opportunities in more investment areas about ESG start-ups, thereby improving its scouting capability. At the same time, the greater the breadth of information sharing, the higher is the information heterogeneity and information complementarity of VC network communities in relation to VC investment projects, which is conducive to resource allocation and utilization for solving conventional or unconventional problems. It also facilitates the allocation of resources for acquiring new investment opportunities, which will improve its utilization of configuration, thereby improving its coaching capability. In summary, this paper proposes the following hypotheses:

**H3.** 
*The breadth of information sharing in the VC network community has a positive impact on the community’s investment capability.*


**H3a.** 
*The breadth of information sharing in the VC network community has a positive impact on the scouting capability.*


**H3b.** 
*The breadth of information sharing in the VC network community has a positive impact on the coaching capability.*


The depth of information sharing affects the authenticity of information and the value of information, thereby affecting the scouting capability. The greater the depth of information sharing is, the more frequent or repeated the communication and sharing between ESG start-ups project information or experience information among community members are, which will strengthen the community’s capability to perceive investment opportunities. For the VC network community, the higher is the depth of information sharing, the more frequent the information sharing among community members, and the more trust that they can generate between each other, which helps to obtain “second-party opinions” about investment ESG start-ups and reduce opportunistic behavior. Cumming [21] believes that “second-party opinions” can be obtained in the VC network, which is the information of other VCs on the start-ups, which is beneficial to reduce the possibility of investing in the “lemon” start-ups. Similarly, Lee [26] holds that the information sharing accumulated through repeated transactions reduces the information asymmetry between participants and reduces the opportunistic behavior of partners.

Information sharing has a deep impact on information communication, information understanding and information reorganization, which affects the coaching capability. According to Levinthal and March [31], on the one hand, the depth of knowledge sources can effectively reduce the mismatch of information at the communication level and reduce the cost of information communication. On the other hand, the depth of knowledge sources can deepen the concept of this information or knowledge and thus is conducive to the development of more valuable information or knowledge, that is, information reorganization. For the VC field, the experience information is particularly professional, and VCs often focus on investing in specific industries and at specific stages to accumulate specialized investment experience. The depth of the sharing of investment experience information can facilitate the reorganization of investment experience and the formation of new investment experience.

Therefore, we believe that the greater the depth of information sharing, the easier it is for the VC network community to obtain more authentic or valuable information, such as “second-party opinions” on ESG start-ups, which can strengthen the VC network community’s ability to grasp the investment opportunities and the perception of partner selection, thereby improving the scouting capability. At the same time, the greater the depth of information sharing, the more the VC network community can reduce the cost of information communication, and the more conducive it is to information reorganization, thereby improving the coaching capability.

**H4.** 
*The depth of information sharing in the VC network community has a positive impact on the community’s investment capability.*


**H4a.** 
*The depth of information sharing in the VC network community has a positive impact on the scouting capability.*


**H4b.** 
*The depth of information sharing in the VC network community has a positive impact on the coaching capability.*


#### 3.4.3. Mediating Role of Community Investment Capability

This paper believes that the breadth of information sharing and the depth of information sharing influence the performance of community investment through the intermediary path of scouting capability and coaching capability. The construction of scouting capability and coaching capability is based on community information sharing. The stronger the community-sensing initiative is, the better the quantity and quality of information about investment opportunities. This is also the reflection of the breadth and depth of information sharing. At the same time, a network community with strong scouting capability can better screen and evaluate ESG start-ups and guarantee the improvement of investment performance in the quality of ESG start-ups. The stronger is the utilization of the configuration, the richer or more professional the resources such as the mastery and the information that can be controlled. Moreover, a network community with strong coaching capability can also mobilize, utilize, and reorganize the information and other resources for value-added services such as supervision and corporate governance, and promote the rapid and sustainable development of the ESG start-ups, ensuring that it has better investment performance.

Therefore, the scouting capability and coaching capability establish a bridge between community information sharing and investment performance. The difference in the breadth and depth of information sharing will result in different scouting capability and coaching capability. The scouting capability and coaching capability directly determine the project screening and value-added service quality of the ESG start-ups, which in turn affects its investment performance. Based on the above analysis, this paper proposes the following hypotheses:

**H5.** 
*Community investment capability plays a mediating role in the relationship between information sharing and investment performance.*


**H5a.** 
*The scouting capability plays a mediating role in the relationship between the breadth of information sharing and investment performance.*


**H5b.** 
*The scouting capability plays a mediating role in the relationship of investment performance in the depth of information sharing.*


**H5c.** 
*The coaching capability plays a mediating role in the investment performance relationship in the breadth of information sharing.*


**H5d.** 
*The coaching capability plays a mediating role in the relationship of investment performance in the depth of information sharing.*


## 4. Research Design

### 4.1. Data Source and Sample

The research data are mainly from the private equity database of Zero2IPO, China’s largest VC data vendor, and are supplemented by the CVsource database of ChinaVenture. We observe the VC data from 2000 to 2017, and select a sample of VC investment in the ESG start-ups between 2003 and 2012. The interval begin in 2003, it mainly to bulid the VC network and network community for three years from 2000 to 2002, and to observe community information sharing within this time window. The five-year time window from 2013 to 2017 is left to observe the lagging performance, so the deadling for 2012 is selected as the sample.

First, this paper downloads the investment events from 2000 to 2012 from the database of Zero2IPO. We build a VC network in a matrix using the three-year moving time window of syndicated investment [9], identify the network community according to the GN algorithm used in Bubna [14], and calculate indicators such as community information sharing and community investment capability. Second, this paper downloads the exit events from 2004 to 2017 from the database of Zero2IPO. We observe whether the VC exit via an IPO or M&A, and calculate the investment performance. Finally, based on the compiled index data, the relationship between the variables is analyzed using Stata software (StataCorp., College Station, TX, USA). The principle of sampling in this paper is that a VC has at least one investment in ESG start-ups during the investment behavior observation period. Last, we collect a total of 8739 rounds of ESG start-ups investment data from 563 VCs from 227 communities. This paper divides the sample into ten groups, as shown in Table 1.

### 4.2. Variable Definitions

#### 4.2.1. Dependent Variable

According to Hochberg et al. [9], this paper measures the performance of community investment using a successful exit ratio. The percentage of successful exits refers to the proportion of the number of events exited by successful exits such as IPOs or mergers in the investment events of all members of the VC network community divided by the total number of investment events. Specifically, we focus on the VC network community in 2000–2002 and observe the investment events of all its community members in 2004. Next, we judge whether their investment events were successfully exit through IPO or M&A in 2005–2009 and then calculate the proportion of successful exit events in the total number of investment events.

#### 4.2.2. Independent Variable

The sharing of community information is characterized by two dimensions: information sharing breadth and information sharing depth. In related research, Awate and Mudambi [32] argues that the breadth of patent innovation portrays the scope of knowledge or the integration of knowledge, and the depth of patent innovation represents the economies of scale of knowledge or the specialization of knowledge. Miranda and Saunders [33] argues that the breadth of information sharing is measured by the number of different discussion topics initiated by group members and the depth of information sharing is measured by the number of times a group member has discussed a topic that has been initiated. Based on existing studies and the availability of database indicators, this paper defines the breadth of community information sharing as the sharing scope of project information or experience information among community members, and it is measured by the number of partners who syndicate investment in members of the community. The depth of information sharing refers to the frequency of the sharing of empirical information about project information among community members, and it is measured by the number of syndicated investment event rounds among members of the community divided by the number of syndicated investment partners.

#### 4.2.3. Mediator Variable

According to Meglio et al. [17], this paper believes that the community’s scouting capability is to use the existing resources of the community to sense and prioritize future investment opportunities. Therefore, this paper uses the data of the previous year to measure the network community resources, that is, the average of industry diversification and geographic diversification of projects invested in by members of the online community in t-1 years. The coaching capability is to use existing resources to better allocate, develop and utilize entrepreneurial start-ups. Therefore, this paper uses the current network community resources to measure the coaching capability, that is, the average of the industry diversification and geographic diversity of the start-ups invested in by members of the online community in year t.

#### 4.2.4. Control Variables

Based on the VC practice and the related research, this paper collects and organizes four categories of variables: the network community level, the VC level, the start-ups level and the macro market level, including community size, community centrality, VC reputation, VC age (logarithmic value), stage of start-ups development, area where the start-ups are located, and exit conditions.

### 4.3. Descriptive Statistics and Relevance Analysis

#### 4.3.1. Direct Effect Test

With regard to Hypothesis 1 and Hypothesis 2, successful exit ratio (*SER*) is the dependent variable, and the information sharing breadth (*ISB*) and information sharing depth (*ISD*) are independent variables, with such parameters as community size (*CS*) and community centrality (*CC*), VC reputation (*VCR*), VC age (*VCA*), start-ups stage (*SS*), start-ups area (*SA*), exit condition (*EC*) being control variables, and control for the fixed effect of the year. We set the model as shown in the following Equation (1).
(1)$$SER=\alpha +\beta_{1}*ISB+\beta_{2}ISD+\beta_{i}controls+\varepsilon$$

For Hypothesis 3 and Hypothesis 4, the scouting capability (*SCA*) and coaching capability (*CCA*) are the dependent variables, and the information sharing breadth (*ISB*) and information sharing depth (*ISD*) are independent variables; the control variables are the same as in 4.1 and control for the fixed effect of the year. We set the model as shown in the following Equation (2).
(2)$$SCA/CCA=\alpha +\beta_{1}*ISB+\beta_{2}ISD+\beta_{i}controls+\varepsilon$$

#### 4.3.2. Mediating Effect Test

In the research hypothesis proposed in this paper, the mediating effect test is Hypothesis 5, which is the mediating effect of testing the community investment capability. According to the stepwise method of Baron and Kenny [34], we set the model as shown in the following Equations (3)–(5).

For Hypothesis 5, the successful exit ratio (*SER*) is the dependent variable, and the information sharing breadth (*ISB*) and information sharing depth (*ISD*) are independent variables; the scouting capability (*SCA*) and coaching capability (*CCA*) are the mediator variables, with community size (*CS*) and community centrality (*CC*), VC reputation (*VCR*), VC age (*VCA*), start-ups stage (*SS*), start-ups area (*SA*), exit condition (*EC*) being control variables, and control for the fixed effect of the year. We set the model as shown in the following Equations (3)–(5).
(3)$$SER=\alpha +\beta_{11}*ISB/ISD+\beta_{1i}controls+\varepsilon$$
(4)$$\mathrm{SCA}/CCA=\alpha +\beta_{21}*ISB/ISD++\beta_{2i}controls+\varepsilon$$
(5)$$SER=\alpha +\beta_{31}*ISB/ISD+\beta_{32}*SCA/CCA+\beta_{3i}controls+\varepsilon$$

To test the mediating effect of the community’s investment capability, the following three conditions must be met:
(1)For Equation (3), if the impact of community information sharing on community investment performance is not significant, that is, $\beta_{11}=0$, there is certainly no intermediary effect. If $\beta_{11}\neq 0$, then we conduct the second step test.(2)For Equation (4), if the impact of community information sharing on community investment capability is not significant, that is, $\beta_{21}=0$, there is certainly no intermediary effect. If $\beta_{21}\neq 0$, then we conduct the third step test.(3)For Equation (5), if the impact of community investment capability on community investment performance is significant, that is, $\beta_{32}\neq 0$ and the impact of community information sharing on community investment performance is not significant, that is $\beta_{31}=0$, the community investment capability is thus fully intermediary. If $\beta_{32}\neq 0$ and, $\beta_{31}\langle \beta_{11}$, thus, the community investment capability is partially intermediary.

### 4.4. Descriptive Statistics and Relevance Analysis

Table 2 and Table 3 are descriptive statistics of the main variables and pearson correlation analysis results, respectively. As we can see from Table 2, the average successful exit ratio of the community in the sample is 0.268, which indicates that 26.8% of the start-ups invested in by the community exit successfully through IPO or M&A, which is higher than the average level of a single VC. The possible explanation is that the VC network community is more conducive to resource aggregation, thereby improving their investment performance [14]. The average value of information sharing breadth is 14.370, the standard deviation is 22.960, and the average value of information sharing depth is 4.138, while the standard deviation is 2.967, which indicates that the difference of information sharing breadth in the community is not largeg, but the difference of information sharing depth is large.

As seen from Table 3, the coefficients of the breadth and depth of information sharing on the successful exit ratio of the community are 0.636 *** and 0.603 ***; the coefficients of the breadth of information sharing on scouting capability and coaching capability are 0.531 *** and 0.449 ***; and the coefficients of the depth of information sharing on scouting capability and coaching capability are 0.597 *** and 0.603 *** respectively. These results show that there is a significant positive correlation between the breadth and depth of information sharing on the successful exit ratio, scouting capability, and coaching capability. However, Pearson correlation analysis does not consider the impact of other control variables, so further analysis and testes are needed.

## 5. Empirical Results

### 5.1. Direct Effect Test

This part empirically examines the influence of information sharing breadth and depth on successful exit ratio, scouting capability, and coaching capability. We use the OLS model for analysis. The regression results are shown in Table 4.

Model 1 maintains only the control variables, the explanatory variable is the successful exit ratio. Model 2 and Model 3 increase the breadth and depth of information sharing variables, respectively, to test their impact on the successful exit ratio. As we see from the results of model 2, we see that the regression coefficient of information sharing breadth is 0.010 and significant at the 1% level; from the results of model 3, we see that the coefficient of information sharing depth is 0.045 and significant at the 1% level. The above results show that the breadth and depth of information sharing significantly affect the successful exit ratio, that is, community information sharing has a significant positive impact on investment performance. Therefore, Hypotheses 1 and 2 are supported.

Model 4 for controls the control variables, and the interpreted variable is scouting capability. Model 5 and Model 6 add information sharing breadth and depth variables on the basis of model 4 to test their influence on scouting capability. From the results of model 5, we see that the coefficient of information sharing breadth is 0.007 and significant at the 1% level; from the results of model 6, we see that the coefficient of information sharing depth is 0.039 and significant at the 1% level. The above results show that the breadth and depth of information sharing have a significant impact on scouting capability, that is, community information sharing has a significant positive impact on scouting capability. Therefore, Hypotheses 3a and 3b are supported.

Model 7 controls for the control variables, and the interpreted variable is coaching capability. Model 8 and model 9 add information sharing breadth and depth variables on the basis of model 7 to test their impact on coaching capability. From the results of model 8, we can see that the coefficient of information sharing breadth is 0.007 and significant at the 1% level; from the results of model 9, we see that the coefficient of information sharing depth is 0.044 and significant at the 1% level. The above results show that the breadth and depth of information sharing significantly affect the coaching capability, that is, the community information sharing has a significant positive impact on the coaching capability. Therefore, Hypotheses 4a and 4b are supported.

### 5.2. Mediation Effect Test

We use the stepwise method to test the intermediary effect of community investment capacity in this part. We examine the intermediary effect of community investment capacity. Specifically, we use the OLS model (ordinary least squares method) to analyze the impact of information sharing breadth and depth, scouting capability and coaching capability on the successful exit ratio. The regression results are shown in Table 5 below.

Model 1 examines the impact of information sharing breadth and scouting capability on the successful exit rate. From the results of Model 1, it is seen that the coefficient of information sharing breadth and scouting capability are 0.008 and 0.258 respectively and both significant at the 1% level. The above results indicate that the breadth of information sharing and the scouting capability significantly affect the proportion of successful exits. Compared to Table 4, it is seen that the coefficient of information sharing breadth is reduced from 0.010 to 0.008, which means the influence of information sharing breadth on the successful exit ratio is significantly reduced under the effect of scouting capability. Therefore, the scouting capability is partially mediated between the breadth of information sharing and the proportion of successful exits, so Hypotheses 5a is supported.

Model 2 examines the impact of information sharing breadth and coaching capability on the successful exit ratio. From the results of Model 2, it is seen that the coefficient of information sharing breadth and coaching capability are 0.008 and 0.278 respectively and both significant at the 1% level. The above results show that the information sharing breadth and coaching capability significantly affect the successful exit rate. Compared to Table 4, it is seen that the coefficient of information sharing breadth is reduced from 0.010 to 0.008, that is, the influence of the information sharing breadth on the successful exit ratio is significantly reduced under the effect of coaching capability. Therefore, coaching capability plays a partial intermediary role between the information sharing breadth and the successful exit ratio, so Hypotheses 5c is supported.

Model 3 examines the impact of information sharing depth and scouting capability on the successful exit rate. From the results of Model 3, it is seen that the coefficient of information sharing depth and scouting capability are 0.030 and 0.384 respectively and both significant at the 1% level. The above results show that the depth of information sharing and the scouting capability significantly affect the proportion of successful exit. Comparing to Table 4, it is seen that the coefficient of information sharing depth is reduced from 0.045 to 0.030, that is, the influence of information sharing depth on the successful exit ratio is significantly reduced under the effect of scouting capability. Therefore, scouting capability plays a partial intermediary role between the depth of information sharing and the proportion of successful exits, so Hypotheses 5b is supported.

Model 4 examines the impact of information sharing depth and coaching capability on the successful exit ratio. From the results of Model 4, it is seen that the coefficient of information sharing depth and coaching capability are 0.029 and 0.377 respectively and both significant at the 1% level. The above results show that the depth of information sharing and the coaching capability significantly affect the proportion of successful exits. Compared to Table 4, it is seen that the coefficient of information sharing breadth is reduced from 0.045 to 0.029, that is, the influence of information sharing depth on the successful exit ratio is significantly reduced under the effect of coaching capability. Therefore, coaching capability plays a partial intermediary role between the information sharing depth and the successful exit ratio, so Hypotheses 5d is supported.

### 5.3. Robustness Test

In order to ensure the robustness of our conclusions, this part test the robustness tests from two aspects.

First, we eliminate outliers. To further ensure the quality of the sample data, this section removes the abnormal values of 10% of the variables separately for re-examination. The results show that the regression coefficients and significance in each model were improved to different extents, and the signs of the regression coefficients did not change. Overall, these tests show that our conclusions above are robust. The results are shown in Table 6 below.

Second, we remeasure the performance. We use financial indicators internal rate of return (IRR) for replacement. Due to the confidentiality and lack of mandatory disclosure of unlisted companies, the IRR data is more difficult to obtain. In the database of Zero2IPO, the IRR is only disclosed by a small number of investment events, so we use a small sample for robustness testing. Overall, these tests show that our conclusions above are robust. The results are shown in Table 6 below.

## 6. Conclusions

This paper studies the impact of community information sharing on investment performance in VC networks and its role path. Based on the theoretical analysis, this paper uses Chinese VC and ESG start-ups data for empirical testing and further discusses the empirical test results. The main conclusions of this study are as follows:

(1) VC network community information sharing directly affects investment performance and has a significant positive impact on investment performance. Specifically, the breadth of information sharing and the depth of information sharing have a significant positive impact on the percentage of successful exits. (2) VC network community information sharing has a significant positive impact on community investment capabilities. Specifically, the breadth of information sharing and the depth of information sharing have a significant positive impact on the scouting capability and coaching capability. (3) VC network community information sharing indirectly affects investment performance, and community investment capability plays a partial intermediary role in its impact relationship. Specifically, scouting capability plays a part role in mediating the influence of information sharing breadth and information sharing depth on investment performance, and coaching capability plays a part role in mediating the influence of information sharing breadth and information sharing depth on investment performance.

However, this study still has some expandable directions. We do not distinguish the types of information sharing, such as project information sharing and experience information sharing for specific discussion; at the same time, VCs with different backgrounds, such as government background, foreign investment background, private Background VC, may have different willingness to share information, and may have different effect on investment performance. These issues are worth exploring the next step.

## Figures and Tables

**Figure 1 ijerph-16-01023-f001:**
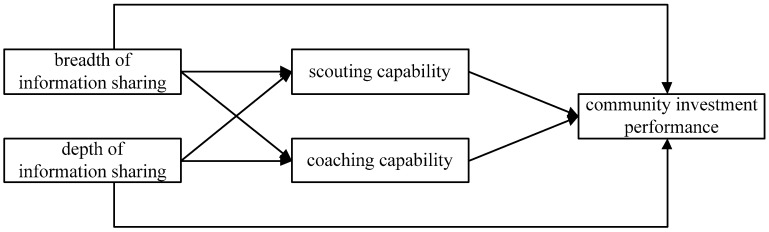
The theoretical model.

**Table 1 ijerph-16-01023-t001:** Sample groups and observation time distribution.

Group	Time Window of VC Community	Time Window of Information Sharing	Time Window of Investment Capability	Time Window of Investment Events	Time Window of Performance
1	2000–2002	2000–2002	2003	2003	2004–2008
2	2001–2003	2001–2003	2004	2004	2005–2009
3	2002–2004	2002–2004	2005	2005	2006–2010
…	…	…	…	…	…
9	2008–2010	2008–2010	2011	2011	2012–2016
10	2009–2011	2009–2011	2012	2012	2013–2017

**Table 2 ijerph-16-01023-t002:** Descriptive Statistics.

Variables	N	Mean	sd	Min	p25	p50	p75	Max
Successful exit ratio(SER)	227	0.268	0.273	0.000	0.000	0.200	0.400	1.000
Breadth of information sharing(ISB)	227	14.370	22.960	0.000	3.000	5.000	14.000	160.000
Depth of information sharing(ISD)	227	4.138	2.967	0.000	2.000	3.125	5.500	12.750
Scouting capability(SCA)	227	0.386	0.249	0.000	0.188	0.327	0.568	1.000
Coaching capability(CCA)	227	0.429	0.250	0.070	0.230	0.360	0.613	1.000
Community size(CS)	227	13.670	26.780	3.000	3.000	6.000	11.000	205.000
Community centrality(CC)	227	0.815	0.703	0.120	0.333	0.570	1.035	3.321
VC reputation(VCR)	227	3.582	6.313	0.000	0.000	1.366	3.901	40.030
VC age(VCA)	227	2.451	0.671	0.693	1.992	2.398	2.725	4.642
Start-ups stage(SS)	227	0.735	0.320	0.000	0.576	0.811	1.000	3.000
Start-ups area(SA)	227	0.467	0.321	0.000	0.200	0.500	0.681	1.000
Exit conditions(EC)	227	45.650	13.370	1.000	41.750	46.530	50.290	112.000

**Table 3 ijerph-16-01023-t003:** Person correlation matrix.

Variables	1	2	3	4	5	6	7	8	9	10	11	12
Successful exit ratio	1.000											
Breadth of information sharing	0.636 ***	1.000										
Depth of information sharing	0.603 ***	0.547 ***	1.000									
Scouting capability	0.611 ***	0.531 ***	0.597 ***	1.000								
Coaching capability	0.599 ***	0.449 ***	0.603 ***	0.718 ***	1.000							
Community size	0.240 ***	0.683 ***	0.276 ***	0.214 ***	0.125 *	1.000						
Community centrality	0.286 ***	0.113 *	0.305 ***	0.362 ***	0.384 ***	0.155 **	1.000					
VC reputation	0.341 ***	0.245 ***	0.305 ***	0.389 ***	0.320 ***	0.162 **	0.275 ***	1.000				
VC age	0.139 **	0.041	0.048	0.026	0.147 **	0.042	0.211 ***	0.123 *	1.000			
Start-ups stage	0.075	0.008	−0.020	0.001	−0.032	−0.005	−0.133 **	0.125 *	−0.091	1.000		
Start-ups area	0.057	0.014	0.012	−0.029	0.041	0.091	0.238 ***	−0.062	0.104	−0.038	1.000	
Exit conditions	0.201 ***	0.215 ***	0.076	0.110 *	0.040	0.047	−0.158**	0.177 ***	0.024	0.003	−0.237 ***	1.000

* Significant at 10%, ** Significant at 5%, *** Significant at 1%.

**Table 4 ijerph-16-01023-t004:** Impact of community information sharing on community investment performance and community investment capability.

Variables	Successful Exit Ratio			Scouting Capability			Coaching Capability		
	Model 1	Model 2	Model 3	Model 4	Model 5	Model 6	Model 7	Model 8	Model 9
Community size	0.002 **	−0.004 ***	0.001	0.001 *	−0.003 ***	0.000	0.001	−0.004 ***	−0.000
	(0.001)	(0.001)	(0.001)	(0.001)	(0.001)	(0.001)	(0.001)	(0.001)	(0.001)
Community centrality	0.069 **	0.077 **	0.025	0.112 ***	0.119 ***	0.075 ***	0.089 ***	0.095 ***	0.045 *
	(0.031)	(0.031)	(0.032)	(0.025)	(0.025)	(0.022)	(0.026)	(0.027)	(0.025)
VC reputation	0.010 ***	0.006	0.006	0.011 ***	0.008 ***	0.007 ***	0.010 ***	0.006 **	0.005 **
	(0.004)	(0.004)	(0.003)	(0.002)	(0.002)	(0.002)	(0.002)	(0.003)	(0.002)
VC age	0.027	0.026	0.035	−0.028	−0.029	−0.022	0.017	0.015	0.024
	(0.029)	(0.019)	(0.022)	(0.028)	(0.025)	(0.026)	(0.028)	(0.022)	(0.023)
Start-ups stage	0.101	0.088 *	0.099 *	0.011	0.002	0.010	0.017	0.007	0.015
	(0.077)	(0.053)	(0.060)	(0.058)	(0.048)	(0.049)	(0.063)	(0.056)	(0.055)
Start-ups area	0.048	0.041	0.063	−0.054	−0.059	−0.041	−0.029	−0.034	−0.014
	(0.061)	(0.041)	(0.050)	(0.056)	(0.045)	(0.049)	(0.054)	(0.045)	(0.046)
Exit conditions	0.005	0.002	0.004 *	0.001	−0.001	0.001	0.001	−0.002	0.000
	(0.004)	(0.002)	(0.003)	(0.002)	(0.001)	(0.002)	(0.003)	(0.001)	(0.002)
Breadth of information sharing		0.010 ***			0.007 ***			0.007 ***	
		(0.001)			(0.001)			(0.001)	
Depth of information sharing			0.045 ***			0.039 ***			0.044 ***
			(0.008)			(0.005)			(0.005)
Year fixed effects	Yes	Yes	Yes	Yes	Yes	Yes	Yes	Yes	Yes
Intercept	−0.186	−0.128	−0.296 ***	0.228 *	0.272 **	0.134	0.250 **	0.294 ***	0.142 *
	(0.124)	(0.092)	(0.107)	(0.124)	(0.114)	(0.124)	(0.102)	(0.083)	(0.086)
N	227	227	227	227	227	227	227	227	227
r^2^	0.278	0.577	0.464	0.278	0.484	0.442	0.229	0.434	0.440
F	7.400	18.162	9.797	10.776	13.396	14.066	7.668	11.871	16.648

Standard errors are reported in parentheses. * Significant at 10%, ** Significant at 5%, *** Significant at 1%.

**Table 5 ijerph-16-01023-t005:** Impact of Community Information Sharing and Community Investment Capabilities on the Investment Performance of Venture Capital Network.

Variables	Successful Exit Ratio			
	Model 1	Model 2	Model 3	Model 4
Community size	−0.003 ***	−0.003 ***	0.001	0.001
	(0.001)	(0.001)	(0.001)	(0.001)
Community centrality	0.047	0.051 *	−0.004	0.008
	(0.030)	(0.029)	(0.031)	(0.030)
VC reputation	0.004	0.004	0.003	0.004
	(0.004)	(0.003)	(0.003)	(0.003)
VC age	0.033 *	0.021	0.043 **	0.026
	(0.019)	(0.018)	(0.021)	(0.021)
Start-ups stage	0.087 *	0.086	0.095 *	0.093
	(0.048)	(0.058)	(0.052)	(0.066)
Start-ups area	0.056	0.051	0.079 *	0.069
	(0.039)	(0.040)	(0.046)	(0.047)
Exit conditions	0.002	0.002	0.004*	0.004 *
	(0.002)	(0.002)	(0.002)	(0.002)
Breadth of information sharing	0.008 ***	0.008 ***		
	(0.001)	(0.001)		
Depth of information sharing			0.030 ***	0.029 ***
			(0.007)	(0.007)
Scouting capability	0.258 ***		0.384 ***	
	(0.067)		(0.072)	
Coaching capability		0.278 ***		0.377 ***
		(0.073)		(0.079)
Year fixed effects	Yes	Yes	Yes	Yes
Intercept	−0.198 **	−0.209 **	−0.347 ***	−0.350 ***
	(0.094)	(0.090)	(0.104)	(0.099)
N	227	227	227	227
r^2^	0.605	0.613	0.532	0.531
F	21.078	25.013	12.086	12.892

Standard errors are reported in parentheses. * Significant at 10%, ** Significant at 5%, *** Significant at 1%.

**Table 6 ijerph-16-01023-t006:** Results of robustness test.

Variables	Successful Exit Ratio (Winsor 10%)			IRR		
	Model 1	Model 2	Model 3	Model 4	Model 5	Model 6
Community size	0.002 **	−0.004 ***	0.001	0.004 ***	0.002	0.003 ***
	(0.001)	(0.001)	(0.001)	(0.001)	(0.001)	(0.001)
Community centrality	0.063 **	0.076 **	0.021	0.067	0.071	0.022
	(0.030)	(0.031)	(0.032)	(0.061)	(0.062)	(0.058)
VC reputation	0.011 ***	0.006	0.006 *	0.001	−0.001	−0.003
	(0.004)	(0.004)	(0.003)	(0.006)	(0.006)	(0.005)
VC age	0.029	0.027	0.037	0.027	0.021	0.028
	(0.030)	(0.019)	(0.023)	(0.049)	(0.048)	(0.048)
Start-ups stage	0.150	0.132 **	0.144 **	−0.125	−0.087	−0.070
	(0.096)	(0.061)	(0.071)	(0.191)	(0.164)	(0.161)
Start-ups area	0.034	0.033	0.051	0.091	0.106	0.166
	(0.063)	(0.042)	(0.051)	(0.124)	(0.124)	(0.117)
Exit conditions	0.004	0.002	0.004	0.003	0.001	0.003
	(0.004)	(0.002)	(0.003)	(0.003)	(0.003)	(0.003)
Breadth of information sharing		0.010 ***			0.004 **	
		(0.001)			(0.002)	
Depth of information sharing			0.045 ***			0.047 ***
			(0.008)			(0.012)
Year fixed effects	Yes	Yes	Yes	Yes	Yes	Yes
Intercept	−0.185	−0.139	−0.299 ***	−0.079	−0.081	−0.253
	(0.126)	(0.094)	(0.107)	(0.296)	(0.283)	(0.291)
N	227	227	227	147	147	147
r^2^	0.284	0.586	0.468	0.318	0.345	0.394
F	7.931	18.670	10.304	4.738	4.064	7.532

Standard errors are reported in parentheses. * Significant at 10%, ** Significant at 5%, *** Significant at 1%.

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
