# Peer review of "Information Sharing and Investment Performance in the Venture Capital Network Community: An Empirical Study of Environmental-Social-Governance Start-Ups"

_ijerph, 2019, doi:10.3390/ijerph16061023_

Round 1

Reviewer 1 Report

The research topic and content are interesting. I have no problem to recommend it to you after some revisions on it. The details are given as the following.

Section 2.4 should be separated as another section: "theoretical framework". The current structure is strange. 

The literature review section is too much. You don't need to cite that many papers. In fact, some of them are slightly related to your research. In fact, if you want, you don't need the section. You can rather add some of its content in introduction section. 

In section 4.1, you should give more details about the data collection. There is in fact only one sentence at the end of the paragraph about the data collection method. It is not clear. It does not promise the quality of data. 

Overall, the English is acceptable. But there are still some errors needed to be fixed. In addition, the description is a little bit too long. If the journal has page limitation requirement, the paper size has room for reduction. 

Author Response

Peer reviewer #1:

Point 1. Section 2.4 should be separated as another section: "theoretical framework". The current structure is strange.

Response: It is our fault that we did not adequately clarify this topic in the previous version. In our revised manuscript, we separate Section 2.4 as a new section and adjust the structure accordingly.

Point 2. The literature review section is too much. You don't need to cite that many papers. In fact, some of them are slightly related to your research. In fact, if you want, you don't need the section. You can rather add some of its content in introduction section.

Response: Thanks for the great comments. We totally agree with you. We recheck the importance and relevance of each literature in this section and rewrite the literature review.

Point 3. In section 4.1, you should give more details about the data collection. There is in fact only one sentence at the end of the paragraph about the data collection method. It is not clear. It does not promise the quality of data.

Response: It is our fault that we did not give more details about the data collection. In the revised manuscript, we add the description of data collection process into Section 4.1 and present the sample distributions in Table 4.1.

Point 4. Overall, the English is acceptable. But there are still some errors needed to be fixed. In addition, the description is a little bit too long. If the journal has page limitation requirement, the paper size has room for reduction.

Response: Thank you for the valuable suggestions. We check the English carefully and correct several errors. Further, we rewrite the Literature Review and delete some loosely related literature. Thus, the paper length is shortened accordingly.

Reviewer 2 Report

This paper selects a sample of ESG start-ups in China to explore the relationship between venture capital network community information sharing and investment performance. It also examines the intermediary role of investment capability in this relationship. It is an interesting paper. It not only helps to reveal the relationship between venture capital network community information sharing and investment performance, but also provides a theoretical basis and managerial insights for improving the investment performance of ESG start-ups.

 However, there are still some minor changes to revise in my opinion.

C1. The empirical work mainly depends on the Zero2IPO database, but the description of the database is not sufficient. Adding more detailed information about the database would be better.

C2. There is some unrelated literature reviewed in the literature review. Please rewrite the literature review to make it easy to read.

C3. In the robustness test section, no concrete result is listed. Please add a table or figure to demonstrate the robustness test result.

It is recommended that this article need to be slightly modified.

Author Response

Peer reviewer #2:

Point1. The empirical work mainly depends on the Zero2IPO database, but the description of the database is not sufficient. Adding more detailed information about the database would be better.

Response: We apologize for not adequately clarifying this description of database in the previous manuscript. In the revised manuscript, we add the description of data collection process into Section 4.1 and present the sample distributions in Table 4.1.

Point 2. There is some unrelated literature reviewed in the literature review. Please rewrite the literature review to make it easy to read.

Response: Thanks for the great comments. We recheck the importance and relevance of each literature in this section and rewrite the literature review.

Point 3. In the robustness test section, no concrete result is listed. Please add a table or figure to demonstrate the robustness test result.

Response: Thank you very much for your professional comment. We revise the previous manuscript according to your comment, and further add Table 5.3 into Section 5.3. In the revised manuscript, we eliminate outliers firstly. To ensure the quality of the sample data, our paper removes the abnormal values of 10% of the variables separately for re-examination secondly. We measure investment performance by using another financial indicator, internal rate of return (IRR), for replacement thirdly.